# White matter lesions and DTI metrics related to various types of dysfunction in cerebral palsy: A meta-analysis and systematic review

Yu Jiang[☯], Gang Liu[☯], Bowen Deng, Xiaoye Li, Jingpei Ren, Yi Zhao, Chuanyu Hu, Lin Xu, Feng Gao*, Xiaohong Mu[ID]*

Dongzhimen Hospital, Beijing University of Chinese Medicine, Beijing, China

☯ These authors contributed equally to this work.
* gaofeng_1220@163.com (FG); muxiaohong2006@126.com (XM)

**Data Availability Statement:** All relevant data are within the manuscript and its Supporting information files.

## Abstract

### Background

Assessing various types of dysfunction in cerebral palsy is a key factor in the treatment and rehabilitation of patients. The objective of this study was to use meta-analysis and systematic review to identify the specific white matter lesions and DTI metrics strongly associated with various types of dysfunction in cerebral palsy.

### Methods

We conducted a literature search of PubMed, Embase, Cochrane Library and Web of Science databases to identify trials published that had evaluated the correlation between DTI metrics in sensorimotor pathways and function scores in cerebral palsy. Correlation coefficient (r) values were extracted for each study, and the extent of r was quantitatively explored. The remaining part of the study was analyzed qualitatively.

### Results

46 studies involving 1458 children with cerebral palsy, were included. 19 articles for Meta-analysis and 27 articles were descriptively analyzed. DTI metrics such as FA、MD in both sensory and motor pathways significantly correlated with various function ratings. In overall motor dysfunction, compared with the CST and PTR, FA of the PLIC correlated more strongly with GMFCS, and showed no significant heterogeneity (r = -1.28, confidence interval [CI]-1.70 to -0.87,I2 = 38.2%,P = 0.233). In upper limb dysfunction, compared with the AHA and MA2, FA of the CST correlated more strongly with BBT, and showed no significant heterogeneity (r = -0.56, confidence interval [CI]-0.78 to -0.34,I2 = 0.0%,P = 0.511). Lower limb dysfunction and other dysfunctions we used qualitative analysis. The qualitative analysis offered a concise overview of each investigation.

**Funding:** The author(s) received no specific funding for this work.

**Competing interests:** The authors have declared that no competing interests exist.

## Conclusions

This study basically identifies the specific white matter lesions corresponding to overall motor dysfunction, upper limb and lower limb motor deficits and other dysfunctions in patients with cerebral palsy, as well as the associated DTI metrics.

## Introduction

Cerebral palsy (CP) is a non-progressive brain injury that occurs due to factors such as premature birth, dystocia, hypoxia, jaundice, and other causes. It is mainly characterized by abnormal posture and movement problems [1]. The most common subtype is spastic cerebral palsy, accounting for approximately 76–80% of cases [2]. It is characterized by motor impairment caused by heightened muscle tone, muscular spasms, and joint contractures. Furthermore, other types of cerebral palsy manifest as sensory abnormalities, impairments in vision, hearing, language, and intellectual development due to damage to different areas of the brain, severely impacting individual growth and development. This severely affects individual growth and development. Evaluating the nature and extent of impairment in individuals with cerebral palsy might assist in devising tailored treatment approaches and facilitating patient rehabilitation.

Magnetic Resonance Imaging (MRI) is a noninvasive diagnostic method commonly used to diagnose cerebral palsy. It may accurately identify the location and severity of the brain damage associated with cerebral palsy. Diffuse white matter injury is a prevalent cause of cerebral palsy, and it is closely associated with mobility difficulties, language function, and cognitive function. Conventional MRI challenges in accurately measuring and analyzing alterations in brain microstructure. Diffusion Tensor Imaging (DTI) is the sole non-invasive technique for studying the in vivo morphological structure of White Matter (WM) fibers. Tracking and evaluating WM fiber tracts can be done based on the diffusion anisotropy of water molecules, which allows for assessing their morphological and structural integrity. DTI primarily utilizes Fractional Anisotropy (FA) as a metric to measure the anisotropy of water diffusion in tissues. FA can provide insights into the connectivity of nerve fiber bundles in various brain areas. Mean Diffusivity (MD) is a frequently utilized DTI metric that measures of intra- and extracellular water diffusion [3]. It is used to evaluate the integrity of brain tissue. The main function of the apparent diffusion coefficient (ADC) value is to quantitatively analyse the dispersion degree of molecules, and this value represents the diffusion efficiency of water molecules in tissues.

Axial Diffusivity (AD), Radial Diffusivity (RD) and apparent fiber density (AFD) can be more specific about the direction and size of water molecule diffusion in tissues. They are reliable pathologic features of cerebral palsy. DTI metric analysis can provide insights into the extent of brain injury, assess the impact of rehabilitation therapy, and inform the development of treatment strategies [4,5].

Furthermore, numerous research have examined the correlation between DTI metrics and impairment of sensorimotor pathways. Several studies have demonstrated a negative correlation between motor function scores and FA of motor pathways, specifically the corticospinal tract (CST) [6–8] and the posterior limb of the internal capsule(PLIC) [9]. Nevertheless, other research has indicated that sensory pathways, such as the posterior thalamic radiation(PTR) [8], correlate similarly with motor function scores. These findings suggest that motor dysfunction may be linked to many pathways or DTI metrics within the same pathway. However, the

specific pathways and DTI metric most strongly associated with motor dysfunction have not been identified. Several research have shown a positive correlation between upper limb dyskinesia and Melbourne assess-ment 2 (MA2) score [10], whereas other studies have found a negative correlation between upper limb dyskinesia and MA2 score [11]. The findings of several investigations exhibit inconsistencies and necessitate consolidation and examination. While there is limited research on other functional impairments such as language and intelligence, it is still necessary to conduct systematic summaries.

Thus, this work examined the relationship between DTI metrics of sensorimotor pathways and different dysfunction scores using meta-analysis and systematic review. Our primary objective was to identify the specific white matter lesions and DTI metrics linked to various types of impairment in individuals with cerebral palsy. Our secondary objective was to summarize commonly used scales for assessing various types of functional impairment in cerebral palsy clinics.

This study was registered on PROSPERO (CRD42024520557).

## Methods

### Search strategy

This study followed the PRISMA Statement. One investigator searched 4 databases: PubMed, Web of Science, Embase, the Cochrane Library. No specified date, age, sex, or language restrictions. We also searched cited references of relevant trial reports and reviews for potentially eligible studies. The search terms we used were "cerebral palsy", "CP", "congenital cerebral palsy", "monoplegic cerebral palsy "or " quadriplegic infantile cerebral palsy and "diffusion magnetic resonance imaging", "diffusion MRI", "diffusion tensor imaging", "DTI" or "tractography." The search was performed using a combination of subject terms and free words, covering the period from database construction to March 2024.

### Study selection

Inclusion criteria in the present meta-analysis and systematic review were as follows: (1) data were acquired from children who had been diagnosed with cerebral palsy by pediatric neurologists; (2) diffusion tensor imaging of the brain had been performed; and (3) the relationship between DTI metrics of sensorimotor pathways and various types of dysfunction scales scores was investigated.

The following studies were excluded: (1) review articles, letters, comments, and case reports; (2) those that provided no relevant data or did not obtain the full data; and (3) studies that reported duplicate patient data.

Following the elimination of duplicates using Endnote software, two investigators conducted a thorough examination of all articles. Initially, they assessed the titles and abstracts and evaluated the complete text to ascertain if they fulfilled the inclusion and exclusion criteria. Disagreements were settled through consultation, directly or involving a third investigator.

### Quality assessment and data extraction

Two investigators independently rated the methodological quality of the included studies using the Newcastle-Ottawa Scale (NOS). Two investigators autonomously extracted information from each study, and cross-checked the extracted information. In instances of dispute, a comprehensive conversation was held to achieve consensus, or a third investigator was consulted to aid in making a decision. The extracted information from each publication comprised the following: author, publication year, country of origin, sample size, patient age at

MRI scan, sensorimotor pathways, kind of dysfunction, relevant scales, DTI metrics, correlation coefficient (r) value, and P value. Data from both the left and right hemispheres were collected for individuals with bilateral cerebral palsy. Data for unilateral cerebral palsy were collected from the same side of the brain.

## Meta-analysis and systematic review

In this meta-analysis, the r values for the correlation between DTI metrics within different sensorimotor pathways (CST, PTR, PLIC, etc) and the function scores were pooled respectively. The r values were extracted for each study, and the extent of r was quantitatively explored. If missing data occurs, make assumptions based on the nature of the missing data. For example, inferences can be made using a random missing data model or a non-random missing data model. We converted Spearman correlation coefficients to Pearson correlation coefficients. Data heterogeneity was analyzed by using the inconsistency index (I2) value. I2 >50% indicated high heterogeneity and a random effects model was used to analyze combined data from the selected studies. A fixed-effect model should be taken when I2 < 50%. All analyses were performed using the software Stata, version 14. P <0.05 was considered statistically significant. We conducted a qualitative descriptive report for studies where meta-analysis could not be performed.

## Results

### Literature search

A total of 1323 articles were retrieved in four databases, and 8 articles were found through other sources such as references. After removing 391 duplicate articles, the remaining 940 articles were screened. Finally, 46 eligible articles were included (Fig 1).

### Data extraction and quality assessment

The included studies involved a total of 1458 patients with cerebral palsy, aged 0–30 years. Thirteen studies explored the correlation between DTI metrics and the GMFCS in cerebral palsy, whereas twenty-six explored the correlation between DTI metrics and upper limb motor function in cerebral palsy. Seven studies explored lower limb dysfunction, while a total of 11 studies focused on other functional impairments, such as swallowing dysfunction, language dysfunction, visual dysfunction, and so on. The quality assessment scores of included studies based on the NOS ranged from 6 to 9 points. (The specific content of the quality assessment is provided in S2 Table). The detailed information regarding the included studies was presented in S3 Table.

### Meta-analysis and systematic review

**Motor function disorder.** A total of 7 studies conducted meta-analyses, all of which evaluated the motor function impairment of cerebral palsy patients using the Gross Motor Function Classification System (GMFCS) grading. The summary r values for the correlation between FA within different sensorimotor pathways and GMFCS are shown in Figs 2–4. Funnel plot of publication bias for studies are shown in S1–S3 Figs. After pooling 5 studies [12–16], the FA in CST correlated with GMFCS (r = -0.43,95% confidence interval [CI] −0.59 to −0.26) and was not markedly heterogeneous(I2 = 7.4%, P = 0.364). After pooling 4 studies [9,13–15], the FA in PTR correlated with GMFCS (r = -0.46,95% confidence interval [CI] −0.71 to −0.21) and was not markedly heterogeneous (I2 = 15.7%, P = 0.313). However, compared with the CST and PTR, the FA in the PLIC most strongly correlated with GMFCS(r =

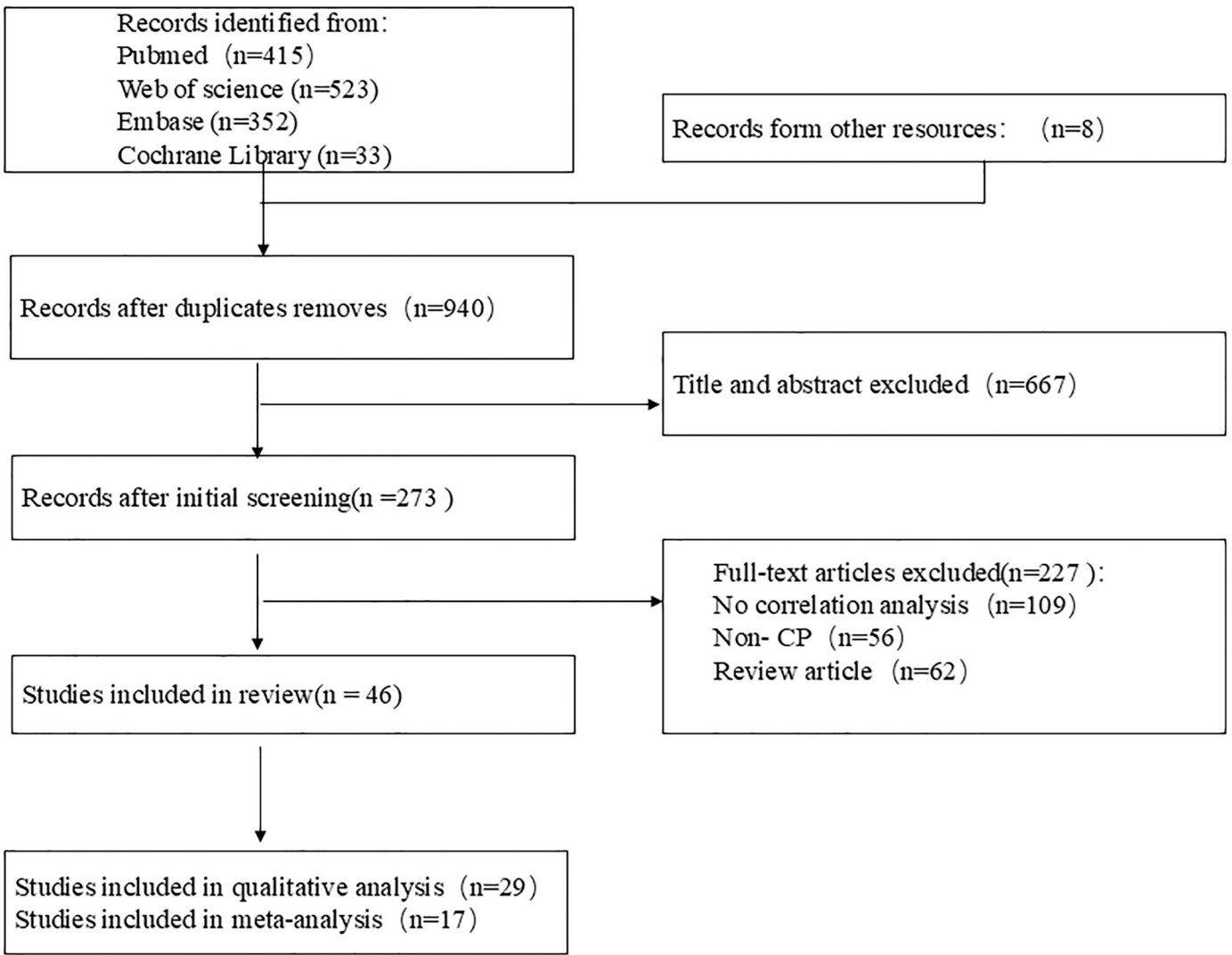

**Fig 1. Flow diagram of study selection.**

-1.28,95% confidence interval [CI] −1.70 to −0.87) after pooling 2 studies [9,16] and was not markedly heterogeneous (I2 = 38.2%, P = 0.233).

There were a total of 6 studies conducted for qualitative analysis. One study [17] found significant correlations between FA in various white matter regions such as the CST, cingulum bundle, superior and inferior longitudinal fasciculi, and frontal-occipital fasciculus, and GMFCS, although specific correlation coefficients (r values) were not reported. Two studies [17,18] utilized special statistical analysis methods, including correlation network analysis and trajectory-based spatial statistics, to confirm the correlation between FA in CST and corpus callosum (CC) in sensorimotor pathways and GMFCS. One study [19] employed the Gross Motor Function Measure (GMFM) as an indicator of motor function disorder, using tract-based spatial statistics to analyze the correlation, and found a significant positive correlation between FA in CST and GMFM. Two studies have analyzed other parameters of DTI besides FA values, and found significant direct correlations between the (ADC) in CST [15], the MD

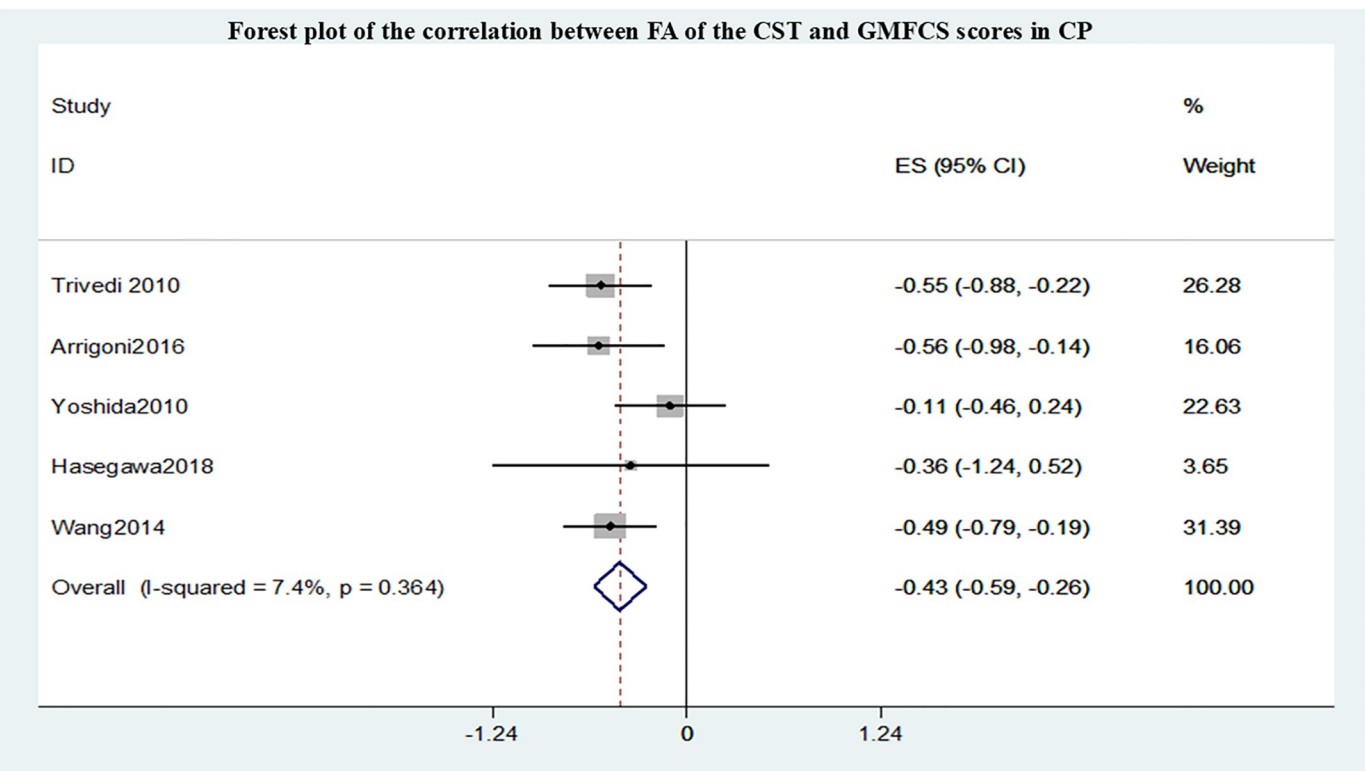

**Fig 2. Forest plot of the correlation between FA of the CST and GMFCS scores in CP.**

in motor tracts 、 the MD of sensory tracts and GMFCS [12]. One study [20] discovered that the number and volume of fibers in the CST were lower in children with low motor function (GMFCS levels 1–3) compared to those with high motor function (levels 4–5) in cerebral palsy.

**Upper limb motor function disorder.** A total of 12 studies conducted meta-analyses, all studied the correlation between FA in CST and upper limb motor function, and the summarized r values were shown in Figs 5–7. Funnel plot of publication bias for studies are shown in S4–S6 Figs. After pooling 3 studies [11,21,22], the FA in CST correlated with Box and Block Test (BBT) (r = -0.56,95% confidence interval [CI] −0.78 to −0.34) and was not markedly heterogeneous(I2 = 0.0%, P = 0.511). After pooling 3 studies [10,11,23], the FA in CST uncorrelated with MA2 (r = -0.43,95% confidence interval [CI] −1.33 to 0.46). After pooling 8 studies [10,23–29], the FA in CST correlated with Assisting Hand Assessment (AHA) (r = 0.46,95% confidence interval [CI] 0.32 to 0.60) and was not markedly heterogeneous (I2 = 14.5%, P = 0.317).

There were a total of 14 studies conducted for qualitative analysis. Three studies utilized the Manual Ability Classification System (MACS) to assess hand function. Children in the low-functioning group (MACS levels IV-V) exhibited more disorganization in the CST on DTI compared to the high-functioning group (levels 1–3) [30]. In the low-functioning group, FA in CST significantly decreased, while fiber number (FN) significantly increased, when compared to the high-functioning group [31]. Moreover, FA in CST, PTR, corona radiata, and superior longitudinal fasciculus were significantly negatively correlated with MACS scores [13]. Seven studies analyzed multiple DTI metrics s of the CST, and found that FA were associated with

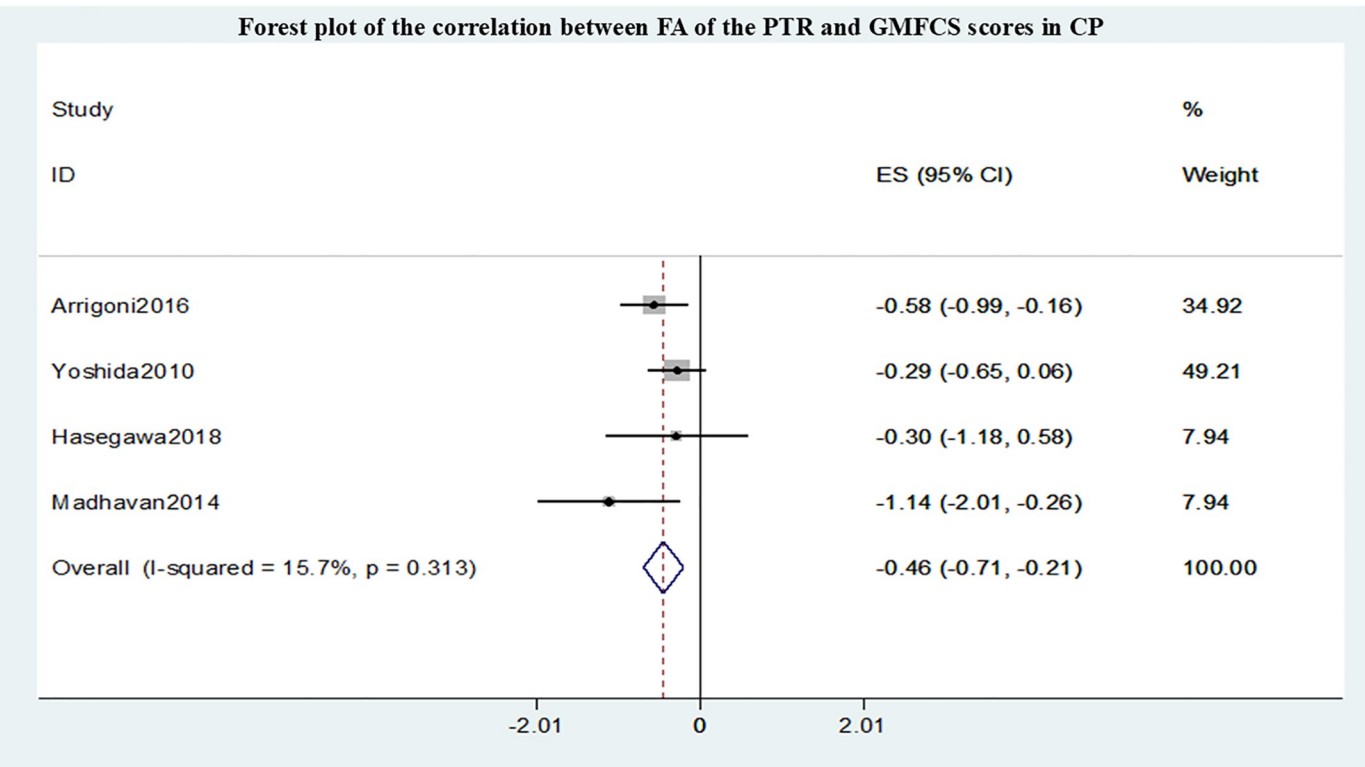

**Fig 3. Forest plot of the correlation between FA of the PTR and GMFCS scores in CP.**

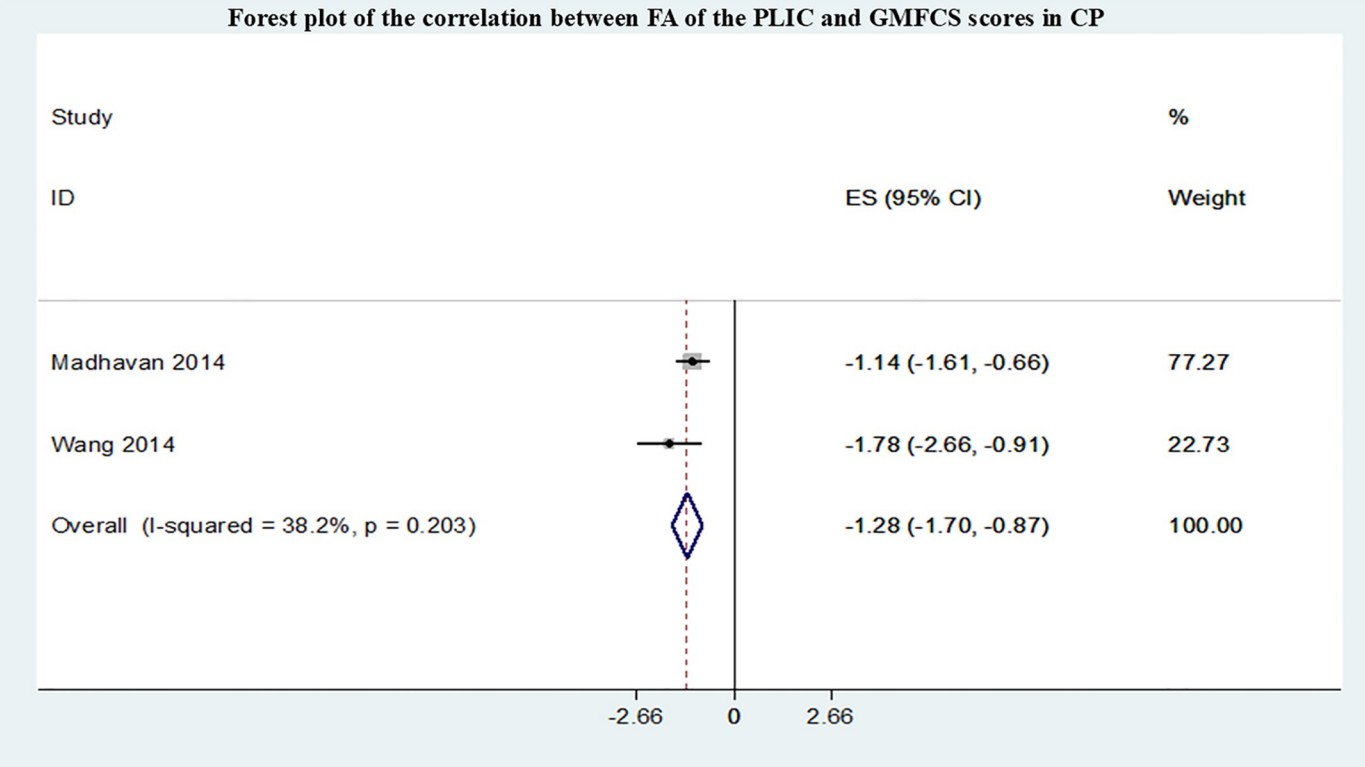

**Fig 4. Forest plot of the correlation between FA of the PLIC and GMFCS scores in CP.**

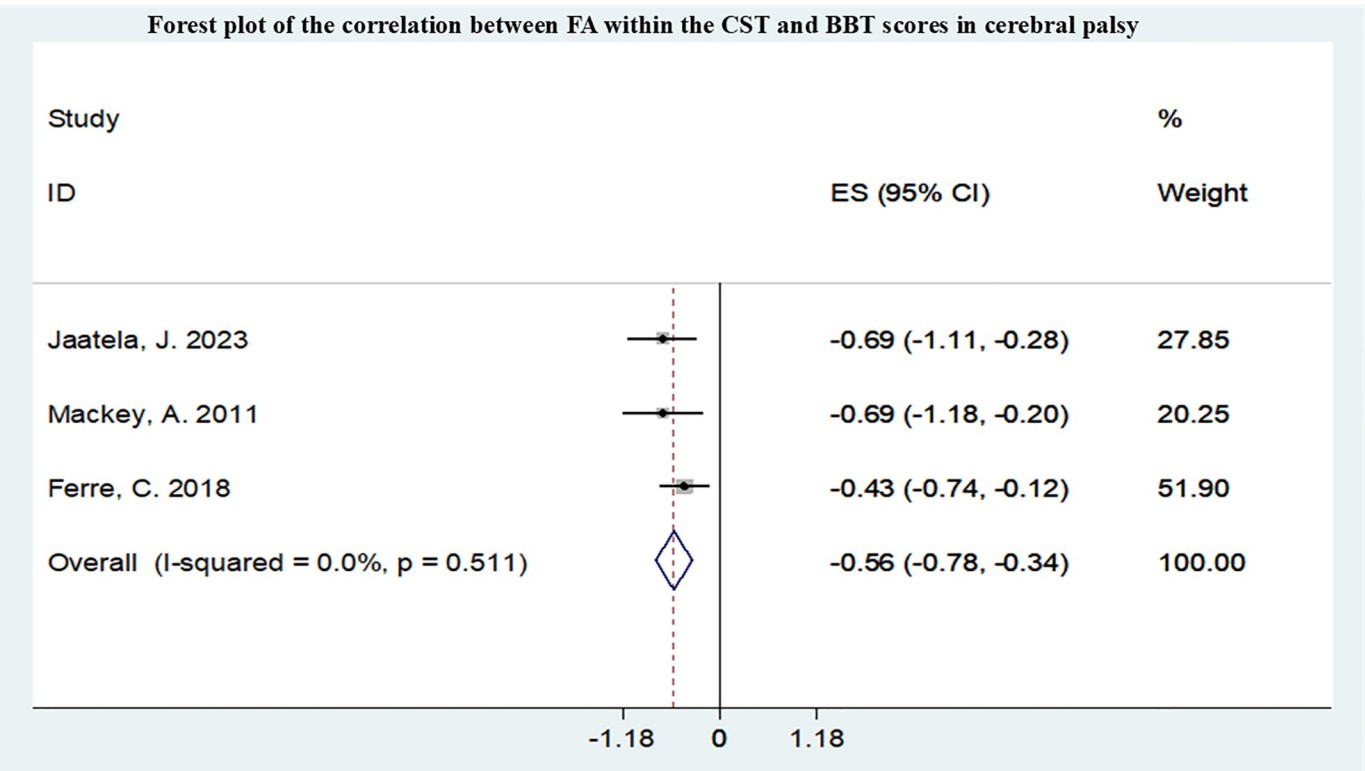

**Fig 5. Forest plot of the correlation between FA within the CST and BBT scores in CP.**

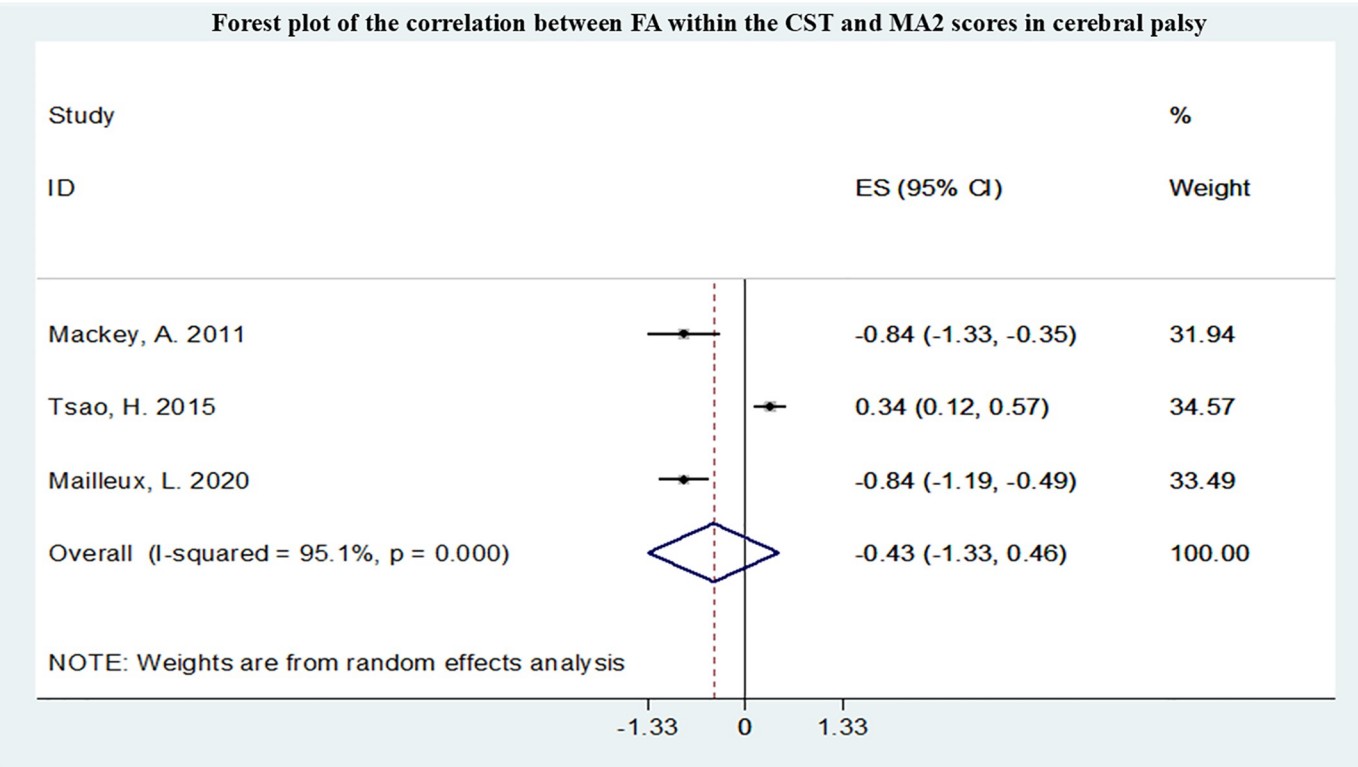

**Fig 6. Forest plot of the correlation between FA within the CST and MA2 scores in CP.**

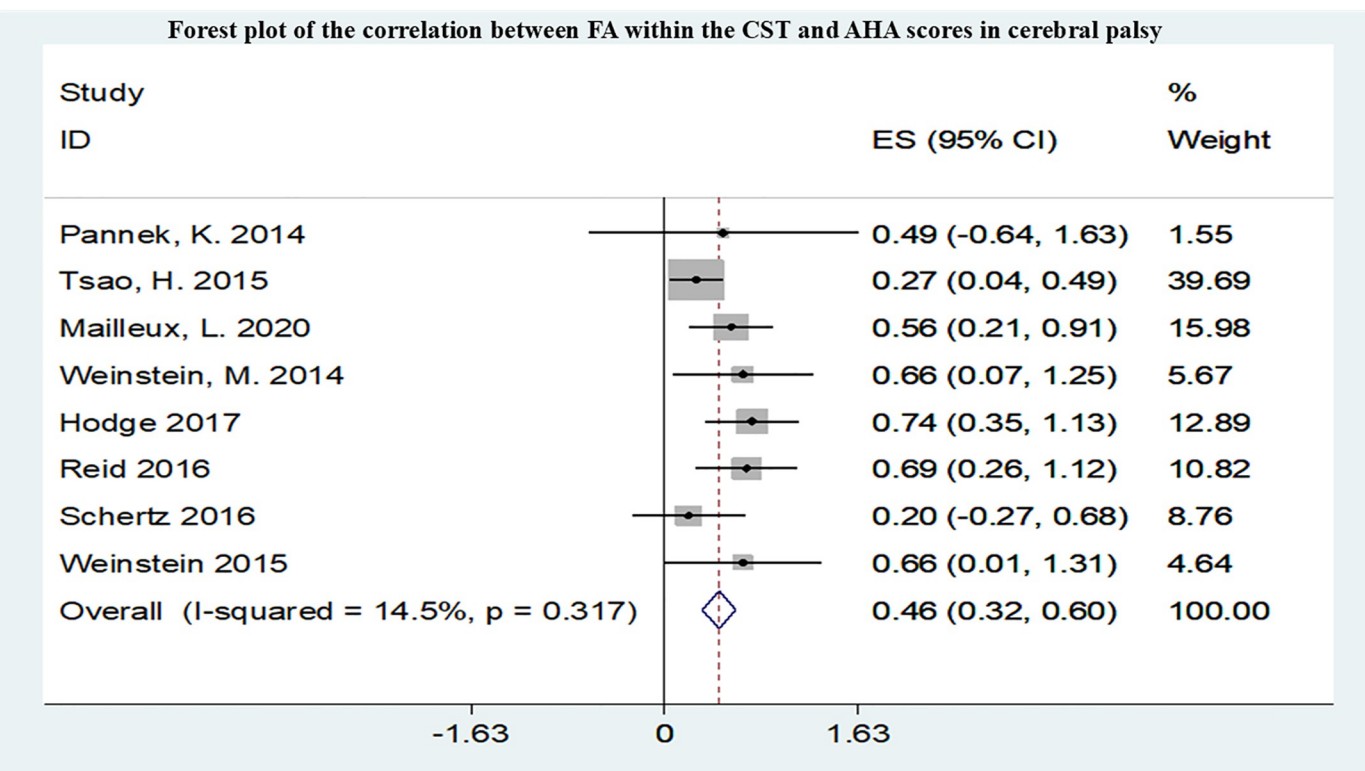

**Fig 7. Forest plot of the correlation between FA within the CST and AHA scores in CP.**

upper limb reaction time [32], BBT scores [33], and Jebsen-Taylor Hand Function Test (JTHFT) scores [34]. MD was correlated with various hand skills [21,23], apparent fiber density (AFD) was positively correlated with hand skills [21], and asymmetry index (AI) and AHA scores also showed significant correlations [35]. FA and MD were found to be correlated with upper limb length in CP patients [36]. Three studies analyzed multiple DTI metrics of the CC and found that AD of the CC was negatively correlated with bilateral coordination scores [37], while AD and MD were positively correlated with JTHFT [38]. MD, FA, and Children's Hand-use Experience Questionnaire (CHEQ) scores showed significant correlations, and FA was significantly negatively correlated with JTHFT scores [25]. One study analyzed the PLIC and found a correlation between FA and CHEQ scores [25]. Additionally, one study utilized Diffusion Tensor Image Analysis along the Perivascular Space (DTI-ALPS) to analyze the glymphatic system, revealing that the ALPS index of the glymphatic system fully mediated the relationship between brain injury burden and MACS levels in children with cerebral palsy, with a mediation effect size of 80% [39].

**Lower limb motor function disorder.** There were a total of 7 studies conducted for qualitative analysis. Four studies analyzed multiple DTI metrics of the CST. One study utilized tract-based spatial statistics to analyze the correlation, revealing a significant positive correlation between FA in CST and the Selective Control Assessment of the Lower Extremity (SCALE) [19]. Walking speed was negatively correlated with MD and RD in CST, and positively correlated with FA in CST. The time Up and Go (TUG) test results were negatively correlated with FA in CST and positively correlated with MD and RD [40]. The FA and AFD of

the non-dominant lower limb bundle in individuals with bilateral paralysis were negatively correlated with lower limb stability [21]. FA and ADC in CST were correlated with the degree of lower limb spasticity. FA in CST were also correlated with GPS scores [41]. Three studies analyzed multiple DTI metrics of the CC, revealing a negative correlation between CC volume and the degree of lower limb spasticity [42]. MD in CC was positively correlated with gait stability indices, but no significant correlation was observed [43]. RD in CC was significantly negatively correlated with SCALE [19]. One study analyzed the posterior thalamic radiation and found a positive correlation between walking impairment and PTR injury scores [44]. One study analyzed the PLIC and found significantly higher scores for lower limb spasticity when PLIC was injured [41].

**Other function disorder.**   There were two studies on **language function disorder**. One study found that in patients with CP who lack any objective language comprehension skills, no language areas could be observed in DTI scans. The language and corticospinal tracts of CP patients were smaller compared to the normal control group [45]. Another study found a negative correlation between language peabody picture vocabulary test-IV (PPVT-IV) scores of CP patients and the average white matter MD in the left temporal lobe, but not in the right temporal lobe. Within the corpus callosum, the FA in the temporal lobe tract were significantly correlated with PPVT-IV scores [46].

There were two studies on cognitive impairment and intellectual developmental disorders. One study found that the FA of different white matter regions such as (parietal white matter (PWM), frontal white matter (FWM), occipital white matter (OWM), temporal white matter (TWM)) were positively correlated with cognitive tests and IQ scores in individuals with spastic diplegia, while MD negatively correlated with scores [47]. Another study found that the white matter regions significantly correlated with FA and IQ scores were more numerous than the regions correlated with MD and IQ [48].

There were three studies on **visual impairment**. One study found that the visual perceptual ability of the SCP is directly proportional to the integrity of the white matter in the right occipital lobe (volume and FA) [49]. Another study discovered that in individuals with unilateral spatial neglect due to visual impairment, the FA of the optic radiation (OR) white matter are correlated with results of visual spatial assessments [50]. A third study found a significant correlation between reduced FA of the superior longitudinal fasciculus (SLF) and various visual tasks in individuals with CP who have visual impairments [51].

There was one study on **sensory abnormality**. The study found that children with abnormal scores in the PTR pathway have higher touch thresholds on the contralateral side of the body. Significant correlations were observed between damage to the right PTR and reduced somatosensory perception in the contralateral upper and lower limbs in individuals with CP [51].

There were two studies on **Executive dysfunction**. One study found a significant correlation between MD in CST on the same side as the lesion in individuals with unilateral cerebral palsy (UCP) and Flanker task performance. Additionally, the connectivity between the anterior cingulate cortex (ACC) and the prefrontal cortex of the contralateral hemisphere was related to Flanker task performance [52]. Another study found a significant correlation between MD and executive function, primarily located in the superior longitudinal fasciculus and superior front occipital fasciculus of the left hemisphere [48].

There was one studies on **swallowing dysfunction**. The study found a correlation between CC integrity and swallowing dysfunction in individuals with UCP, particularly when there is a lesion in the left hemisphere. The DTI metrics of the CC, including decreased FA and fiber count reduction, as well as increased RD, were associated with increased difficulty in swallowing as indicated by Dysphagia Disorder Survey (DDS) scores [53].

## Discussion

The results of this study suggest that the combined effects of damage in multiple white matter regions may lead to motor impairments in patients with cerebral palsy. Through systematic review and evaluation, it has been found that in cerebral palsy, various sensory-motor pathways such as CST, PTR, and PLIC's FA were significantly correlated with overall motor function scores, such as the GMFCS score, without significant heterogeneity observed. Among these, the lesions in the PLIC were most closely associated with overall motor dysfunction. Additionally, other regions of the brain such as the CC, cingulum, and superior and inferior longitudinal fasciculi also showed correlations with GMFCS, although they were not included in the meta-analysis. Clinically, GMFCS is a set of objective evaluation system designed for comprehensive assessment of motor function in cerebral palsy patients, and it is the most used classification system, ranging from level I to V, where higher levels indicate poorer functional abilities. In DTI, FA is a commonly used objective indicator for assessing the extent of brain white matter damage. Previous research has predominantly focused on the association between motor impairments in spastic cerebral palsy and reduced FA values. However, cerebral palsy includes various types, such as mixed cerebral palsy and dyskinetic cerebral palsy. This study comprehensively included all types of cerebral palsy during the search and still found the same conclusion. Therefore, FA can be considered as one of the indicators for predicting motor impairments in patients with cerebral palsy. In DTI, ADC can be used to quantitatively assess the directionality and extent of water molecule movement in bodily tissues and has been found to be related to GMFCS. However, currently, there is only one study on the predictive value of ADC, and further development in this area is needed.

Upper limb movements are finer and more complex, and there are also many scales available for assessing upper limb movements. The results of this study showed that upper limb motor impairments were correlated with damage to the CST and had the highest correlation with the BBT score, but the most studies included in the research applied the AHA to assess upper limb function. The BBT is a simple test method for assessing hand function flexibility, while the AHA is commonly used clinically to assess fine upper limb activities. This study found that the FA in the CST is not correlated with MA2. MA2 is a comprehensive scale for assessing upper limb function in children with neurological disorders. This result is contrary to previous studies, and therefore, increasing the sample size for further investigation is needed. Therefore, there is still a need in clinical practice to further explore comprehensive scales for evaluating upper limb function in cerebral palsy, covering various aspects of upper limb movement function as much as possible. In this study, there were more studies on the correlation between MD and hand function, but meta-analysis was not conducted. FA and MD may be used to predict upper limb function in cerebral palsy. Similarly, lower limb motor dysfunction was mainly associated with damage to the CST and CC, and FA and MD may be used to predict lower limb function in cerebral palsy. There are fewer studies included on other functional impairments. The occurrence of specific functional impairments in cerebral palsy is related to damage to specific small areas of white matter in the brain, such as language impairments associated with the language bundle and the left temporal lobe; visual impairments associated with damage to the optic radiation; sensory abnormalities associated with damage to the thalamic radiation; and swallowing difficulties associated with damage to the corpus callosum. In this meta-analysis and systematic review, most studies have focused on the relationship between the FA in CST and functional impairments. The CST governs voluntary movements of the trunk and limbs and is the primary motor pathway, making it closely associated with motor impairments in patients with cerebral palsy resulting from brain injury [54]. Lesions in the CST can disrupt

cortical motor circuits involved in motor execution and have been shown to be associated with spasticity [55]. The prognosis of motor outcomes in cerebral palsy patients is also closely related to the extent of CST damage, and the recovery of damaged CST correlates with motor function. Evaluating the status of the corticospinal tract is crucial in the field of cerebral palsy rehabilitation [56]. Some studies have demonstrated that damage to the thalamic radiation connecting the thalamus to the occipital and parietal cortices is also associated with multiple functional impairments in children with cerebral palsy. Damage to the PTR reduces connectivity in the sensorimotor cortex, which can weaken the corticospinal descending pathway, leading to functional impairments in cerebral palsy [57,58]. Additionally, the role of the PTR in visual spatial performance further exacerbates functional impairments in cerebral palsy patients, impairing motor coordination [59]. Damage to the corpus callosum is also a key focus of research on cerebral palsy-related functional impairments. The CC is the largest white matter fiber structure in the brain, serving as a vital pathway connecting the left and right brain hemispheres. In children with CP, the less affected hemisphere of the brain attempts to compensate for deficiencies in the more affected hemisphere due to the relationship between the two hemispheres. Therefore, assessing the extent of damage to the CC, which connects the left and right brain hemispheres [60], is crucial for tasks requiring communication between brain hemispheres, especially for bilateral coordination and flexibility in hand movements [61]. However, compared to other sensory-motor pathways, this study indicates that the strongest correlation with motor function scores lies with the PLIC. Anatomically, the PLIC, which the corticospinal fibers traverse, is situated between the thalamus and the lentiform nucleus of the basal ganglia [62]. The PLIC is an essential component of the brain's motor conduction circuit and is a critical anatomical site for evaluating CST damage. Following periventricular leukomalacia (PVL) injury leading to damage in the CST-PLIC segment due to Wallerian degeneration, reduced PLIC-FA represents disruption of white matter fiber integrity, resulting in abnormal neural fiber connections and interruption of neural information transmission pathways, ultimately leading to functional impairments in patients [63,64].

This meta-analysis and systematic review had several limitations. Firstly, the number of studies included in the meta-analysis was limited, and future research could benefit from larger sample sizes. Secondly, most studies have focused on analyzing the correlation between FA in CST and functional impairment scores, with fewer studies investigating other pathways and DTI metrics, making it difficult to conduct further meta-analysis at this time. Thirdly, the majority of included studies were retrospective and cross-sectional in nature. In future research, more prospective studies could be conducted to improve the accuracy of predicting functional impairments in cerebral palsy.

In conclusion, despite the limitations of this meta-analysis and systematic review, its clinical significance is commendable. It identifies the areas of brain white matter damage associated with motor dysfunction, upper and lower limb motor impairments, and other functional impairments in cerebral palsy patients, along with determining the corresponding DTI metrics. These parameters can serve as potential biomarkers for assessing the severity of functional impairments in cerebral palsy and could also be used as a basis for early prediction of cerebral palsy-related dysfunction.

## Supporting information

**S1 Fig. Funnel plot of publication bias for studies on the correlation between FA in CST and GMFCS.**
(TIF)

**S2 Fig. Funnel plot of publication bias for studies on the correlation between FA in PTR and GMFCS.**
(TIF)

**S3 Fig. Funnel plot of publication bias for studies on the correlation between FA in PLIC and GMFCS.**
(TIF)

**S4 Fig. Funnel plot of publication bias for studies on the correlation between FA in CST and BBT.**
(TIF)

**S5 Fig. Funnel plot of publication bias for studies on the correlation between FA in CST and MA2.**
(TIF)

**S6 Fig. Funnel plot of publication bias for studies on the correlation between FA in CST and AHA.**
(TIF)

**S1 Table. Abbreviation.**
(DOCX)

**S2 Table. The NOS evaluation score form of all studies.**
(DOCX)

**S3 Table. Characteristics of the included studies.**
(DOCX)

**S4 Table. Table of all study identifiers.**
(DOCX)

## Acknowledgments

Thank you to all the authors for their contributions to this paper.

## Author Contributions

**Conceptualization:** Xiaohong Mu.

**Data curation:** Xiaohong Mu.

**Formal analysis:** Yu Jiang.

**Methodology:** Jingpei Ren, Yi Zhao.

**Project administration:** Gang Liu, Bowen Deng.

**Resources:** Xiaoye Li, Chuanyu Hu, Lin Xu, Feng Gao.

**Software:** Xiaoye Li, Chuanyu Hu.

**Supervision:** Yi Zhao.

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
