## [Decision Letter · Decision Letter 0]

22 Jul 2024

PONE-D-24-17895White matter lesions and DTI metrics related to various types of dysfunction in cerebral palsy: A meta-analysis and systematic reviewPLOS ONE

Dear Dr. Jiang,

Thank you for submitting your manuscript to PLOS ONE. After careful consideration, we feel that it has merit but does not fully meet PLOS ONE’s publication criteria as it currently stands. Therefore, we invite you to submit a revised version of the manuscript that addresses the points raised during the review process.

We look forward to receiving your revised manuscript.

Kind regards,

Christos Papadelis, Ph.D.

Academic Editor

PLOS ONE

Journal Requirements:

2. In this instance it seems there may be acceptable restrictions in place that prevent the public sharing of your minimal data. However, in line with our goal of ensuring long-term data availability to all interested researchers, PLOS’ Data Policy states that authors cannot be the sole named individuals responsible for ensuring data access (http://journals.plos.org/plosone/s/data-availability#loc-acceptable-data-sharing-methods).

4. Please ensure that you include a title page within your main document. You should list all authors and all affiliations as per our author instructions and clearly indicate the corresponding author.

Reviewers' comments:

Reviewer's Responses to Questions

**Comments to the Author**

1. Is the manuscript technically sound, and do the data support the conclusions?

Reviewer #1: Yes

Reviewer #2: Partly

2. Has the statistical analysis been performed appropriately and rigorously? 

Reviewer #1: Yes

Reviewer #2: Yes

3. Have the authors made all data underlying the findings in their manuscript fully available?

Reviewer #1: Yes

Reviewer #2: Yes

4. Is the manuscript presented in an intelligible fashion and written in standard English?

Reviewer #1: Yes

Reviewer #2: Yes

5. Review Comments to the Author

Reviewer #1: The meta-analysis and review manuscript by Jiang et al. assesses the correlation between various types of disfunction and DTI metrics in cerebral palsy. The manuscript is well written and it is technically sound. There are only a few minor modifications required:

1. There are some instances that the full version of the abbreviations are not mentioned in the text or table (e.g., GPS, ADC, etc.). In addition, there are some discrepancies in the abbreviations and the metric names. For instance, AFD is the abbreviation for 'apparent fiber density'; whereas it is noted as axial diffusivity in the text. You can confirm it with the reference (ref. 20). I suggest adding a table explaining the abbreviations for scales, pathways and DTI metrics and proofreading all the shortened forms in the text and table.

2. There are several typos in table 1. For example:

- 'country' in the first row is not spelled correctly

- non-English font in DTI metrics

- countries 'Spanish', 'Indian', and 'Iranian' should change to 'Spain', 'India', and 'Iran'.

3. There are studies in the field from established research groups which are not cited in the manuscript. For example, the DTI studies of Papadelis group have been omitted from this review paper.  

Reviewer #2: The article has some novelty and positive results. Overall the methodology is sound and the conclusions are supported by the results. However, the details of this paper should be improved. See below for some comments in the different sections of the paper:

1.Abstract：The relatively r values,95% CIs and I² value should be given in main results. The parts of results section seems a little simple.

2.Introduction：

（1）Consider more about the primary and secondary objective of this systematic review and emphasize it.

（2）When mentioned DTI indexes and metric analysis,all the recommended names and clinical significance in the following paper should be involved,for example ADC is missed.

3.Methods：

（1）The search term strategy was not comprehensive (cerebral palsy was used only in cerebral palsy, and no other free words were used).Please improve your search strategy.

（2）Bias was not assessed.Please provide a funnel plot for publication bias in Appendix.

（3）A fixed-effect model should be taken when I2 < 50%，it is not mentioned in this article.

4.Results：

（1）Please list the NOS evaluation score form of all studies.

（2）Upper limb motor function disorder：After pooling 3 studies[9, 10, 22], the FA in CST correlated with MA2 (r =-0.43,95% confidence interval [CI] –1.33 to 0.46) and was markedly heterogeneous （I2=95.1%，P=0.000）. The 95% confidence interval [CI] contains 0,so it's not correlated between FA in CST and MA2.

（3）Upper limb motor function disorder：After pooling 8 studies[9, 22-28], the FA in CST correlated with Assisting Hand Assessment（AHA） (r =0.46,95% confidence interval [CI] 0.32 to 0.60) and was markedly heterogeneous （I2=14.5%，P=0.317）. I2＜50%，P＞0.05，it is not markedly heterogeneous.

5.Discussion：No comments.

6.Overall:General writing quality should be ameliorated.

6. PLOS authors have the option to publish the peer review history of their article (what does this mean?). If published, this will include your full peer review and any attached files.

Reviewer #1: No

Reviewer #2: No

---

## [Author Response · Author response to Decision Letter 0]

29 Jul 2024

I have explained it in the submitted document 'Response to Reviewers'.

---

## [Decision Letter · Decision Letter 1]

18 Sep 2024

PONE-D-24-17895R1White matter lesions and DTI metrics related to various types of dysfunction in cerebral palsy: A meta-analysis and systematic reviewPLOS ONE

Dear Dr. Jiang,

Thank you for submitting your manuscript to PLOS ONE. After careful consideration, we feel that it has merit but does not fully meet PLOS ONE’s publication criteria as it currently stands. Therefore, we invite you to submit a revised version of the manuscript that addresses the points raised during the review process.

We look forward to receiving your revised manuscript.

Kind regards,

Maryam Bemanalizadeh

Academic Editor

PLOS ONE

Journal Requirements:

Reviewers' comments:

Reviewer's Responses to Questions

**Comments to the Author**

1. If the authors have adequately addressed your comments raised in a previous round of review and you feel that this manuscript is now acceptable for publication, you may indicate that here to bypass the “Comments to the Author” section, enter your conflict of interest statement in the “Confidential to Editor” section, and submit your "Accept" recommendation.

Reviewer #1: All comments have been addressed

Reviewer #2: All comments have been addressed

2. Is the manuscript technically sound, and do the data support the conclusions?

Reviewer #1: Yes

Reviewer #2: Yes

3. Has the statistical analysis been performed appropriately and rigorously? 

Reviewer #1: Yes

Reviewer #2: Yes

4. Have the authors made all data underlying the findings in their manuscript fully available?

Reviewer #1: Yes

Reviewer #2: Yes

5. Is the manuscript presented in an intelligible fashion and written in standard English?

Reviewer #1: Yes

Reviewer #2: Yes

6. Review Comments to the Author

Reviewer #1: (No Response)

Reviewer #2: Thanks for your specific revision.Now there is still a little minor details should be revised:

（1）A minor error exists in part“Literature Search”：After removing 391 duplicate articles, the remaining 940（not 930） articles were screened.

（2）Table 1：Adjust the form to three-wire watch and add the corresponding NOS score in the last column.

（3）After adding the corresponding NOS score to table 1，table S2（the NOS Evaluation Score Form of All Studies）and table S1（Abbreviation）in appendix can choosed to be deleted,else there are too many tables.（or you can discuss it with editor eventually）

7. PLOS authors have the option to publish the peer review history of their article (what does this mean?). If published, this will include your full peer review and any attached files.

Reviewer #1: No

Reviewer #2: No

---

## [Author Response · Author response to Decision Letter 1]

24 Sep 2024

I have completed the revisions according to the reviewers' new suggestions and have prepared a document. Please kindly review it.

---

## [Editor Report · Decision Letter 2]

7 Oct 2024

White matter lesions and DTI metrics related to various types of dysfunction in cerebral palsy: A meta-analysis and systematic review

PONE-D-24-17895R2

Dear Dr. Jiang,

We’re pleased to inform you that your manuscript has been judged scientifically suitable for publication and will be formally accepted for publication once it meets all outstanding technical requirements.

Kind regards,

Maryam Bemanalizadeh

Academic Editor

PLOS ONE

---

## [Editor Report · Acceptance letter]

17 Oct 2024

PONE-D-24-17895R2 

PLOS ONE

Dear Dr. Jiang, 

I'm pleased to inform you that your manuscript has been deemed suitable for publication in PLOS ONE. Congratulations! Your manuscript is now being handed over to our production team.

Kind regards, 

on behalf of

Dr. Maryam Bemanalizadeh 

Academic Editor

PLOS ONE